# Effect of Posterior Pelvic Tilt Taping on Pelvic Inclination, Muscle Strength, and Gait Ability in Stroke Patients: A Randomized Controlled Study

**DOI:** 10.3390/jcm10112381

**Published:** 2021-05-28

**Authors:** Tae-sung In, Jin-hwa Jung, May Kim, Kyoung-sim Jung, Hwi-young Cho

**Affiliations:** 1Department of Physical Therapy, Gimcheon University, Gimcheon 39528, Korea; 20160072@gimcheon.ac.kr; 2Department of Occupational Therapy, Semyung University, Jecheon 27136, Korea; otsalt@semyung.ac.kr; 3Department of Physical Education, College of Education, Korea University, Seoul 02841, Korea; kimmay@korea.ac.kr; 4Department of Physical Therapy, College of Health Science, Gachon University, Incheon 21936, Korea

**Keywords:** pelvic inclination, taping, gait, stroke

## Abstract

Objective: Pelvic alignment asymmetry in stroke patients negatively affects postural control ability. This study aimed to investigate the effect of posterior pelvic tilt taping on pelvic inclination, muscle strength, and gait ability in stroke patients. Methods: Forty stroke patients were recruited and randomly divided into the following two groups: the posterior pelvic tilt taping (PPTT) group (*n* = 20) and the control group (*n* = 20). All participants underwent sitting-to-standing, indoor walking, and stair walking training (30 min per day, 5 days per week, for 6 weeks). The PPTT group applied posterior pelvic tilt taping during the training period, while the control group did not receive a tape intervention. Pelvic inclination was measured using a palpation meter (PALM). A hand-held dynamometer and the 10-meter walk test were used to measure muscle strength and gait ability. Results: Significantly greater improvements in the pelvic anterior tilt were observed in the PPTT group than in the control group (*p* < 0.05). Muscle strength in the PPTT group was significantly increased compared to the control group (*p* < 0.05). Significantly greater improvements in gait speed were observed in the PPTT group than the control group. Conclusions: According to our results, posterior pelvic tilt taping may be used to improve the anterior pelvic inclination, muscle strength, and gait ability in stroke patients.

## 1. Introduction

The pelvis is an important structure that connects the torso and lower limbs to support and transmit weight to the lower limbs when performing various functional movements. In addition, the pelvis is a part of the lower trunk in the sitting position but becomes a functional element of the lower limb when standing or walking [1]. Therefore, changes in pelvic alignment in a standing position also affect balance, gait, and functional performance [2]. In addition, proper pelvic control is essential for the establishment of more economical movements and gait [3], and if this is not properly controlled during gait, the speed, stability, and efficiency decrease [4,5,6]. 

In order to properly perform functional movements, such as sitting-to-standing [7] or walking [8], the ability to move weight to the affected lower limb must be preceded. However, in stroke patients, body deficits such as loss of sensation, impaired motor function of the upper and lower extremities, spasticity, and muscle weakness are caused by damage to the blood vessels in the brain. This results in a secondary damage in body control and a change in pelvic alignment, resulting in decreased weight support on the paralyzed side [9]. In addition, stroke patients show a forward tilted posture alignment compared to a healthy person when maintaining a standing posture [10] or show a posture in which the pelvis is tilted anteriorly and to the affected side [9].

Kinesio taping was introduced by Kenzo Kase as a method of attaching an elastic adhesive tape with elasticity similar to that of the skin and is currently used in rehabilitation for various purposes [11]. Taping is applied in combination with other therapeutic techniques to strengthen weakened muscles, regulate joint instability, assist postural alignment, reduce pain, improve blood flow and lymphatic circulation, relieve spasticity, and strengthen muscle function [12,13,14]. Additionally, studies have reported that pelvic inclination decreases when taping is applied for posture correction [15,16,17,18,19]. Lee et al. reported that a significant difference occurred in the pelvic inclination angle when seated workers were divided into a group in which anterior pelvic tilt taping was applied and a group in which it was not, and a slumped sitting posture was maintained for 30 min [16]. Therefore, it was determined that taping can help prevent the musculoskeletal problems caused by an awkward sitting posture [16]. In a study in which posterior pelvic tilt taping was applied for a week to patients experiencing lower-back pain with hyperlordosis, it was found that lumbar lordosis, pain, disability, and abdominal thickness were improved compared to the control group [18]. They said that the skin irritation due to the tape would have improved the abdominal strength by allowing more units of exercise to be mobilized [18]. These studies were also conducted in stroke patients. Mehta et al. reported that when taping was attached to the thoracic and abdomen of stroke patients, the pelvic obliquity and anterior pelvic tilt in a sitting position were improved compared to the control group, which resulted in improved balance [20]. In addition, one study reported that posterior pelvic tilt taping significantly improved pelvic anterior tilt, gait speed, and step length in stroke patients, but further studies are needed as the long-term effects have not been confirmed [21].

Therefore, this study aimed to investigate the effect of posterior pelvic tilt taping on pelvic inclination, lower extremity muscle strength, and gait ability in stroke patients.

## 2. Experimental Section

### 2.1. Participants

We used G*power 3.1.9.2 software (Heinrich-Heine-University Düsseldorf, version 3.1.9.4, Düsseldorf, Germany) to calculate the sample size. In the present study, the mean power was set to 0.8 and the alpha error was set to 0.05. In addition, the effect size was set to 0.8148 based on a pilot study (10 subjects). The analysis of G*power software showed that at least 18 participants would make an acceptable group sample size for each group; thus, 42 participants were recruited in consideration of dropout. 

Participants were recruited from the H rehabilitation centers in Gyeonggi-do. The inclusion criteria were as follows: (1) first episode of unilateral stroke with hemiparalysis caused by hemicerebrum damage; (2) an anterior pelvic inclination greater than 15° (normal range, 11 ± 4°) [22,23,24]; (3) the ability to understand and follow verbal commands; (4) the ability to independently walk for at least 15 m without assistance; and (5) Brunnstrom stage 3 or higher motor recovery of the lower extremity. The exclusion criteria included the following: (1) hemianopia, dizziness, or other symptoms indicating vestibular impairment; (2) neglect and sensory loss; and (3) an orthopedic disease influencing gait. 

This trial was approved by the Gachon University Institutional Review Board (1044396-202006-HR-112-01) and was registered (WHO International Clinical Trials Registry Platform, KCT0005215). Prior to enrollment in the study, the content of the study was explained to all of the participants and written consent was obtained. Table 1 shows the characteristics of the subjects in the posterior pelvic tilt taping (PPTT) and control groups. The subjects were randomly assigned to the PPTT group (*n* = 21) or control group (*n* = 21) using a selection envelope. Before the post-evaluation, one person from each group dropped out due to skin redness and a change of address. A total of 40 subjects were evaluated for pelvic inclination, muscle strength, and gait ability after 6 weeks of training (Figure 1).

### 2.2. Intervention

In both the PPTT group and the control group, sit-to-stand, indoor walking, and stair walking training were performed for 30 min per day, 5 times per week, for 6 weeks. In sit-to-stand training, to increase the weight support for the affected side, the big toe of the affected foot was placed in the middle of the healthy foot, and the subjects were then instructed to stand up without supporting their arm. Indoor walking training included straight walking and S-shaped walking; for stair walking, the subjects were instructed to go up and down stairs that had a height of 10 cm each. All of the training sessions were supervised by a physiotherapist with more than 5 years of rehabilitation experience. In addition, the PPTT group applied posterior pelvic tilt taping. For mechanical correction, a 5-centimeter-wide tape was extended to 50–75% of its original length and attached to both rectus abdominis (RA) and external oblique (EO) muscles [18]. First, the pelvis was tilted posteriorly in the side-lying position and then attached to the EO muscle from the inguinal region to the spinous process of T12. For mechanical correction of the posterior pelvic tilt, the pelvis from the anterior superior iliac spine (ASIS) to the posterior superior iliac spine (PSIS) was attached while tilting in the posterior direction. Finally, the RA muscle was attached from the pubic symphysis to the xiphoid process in the knee bent and supine positions (Figure 2). The tape was changed once every 3 days.

### 2.3. Outcome Measurements

Pelvic inclination was measured using an inclinometer and a palpation meter (PALM; Performance Attainment Associations, St. Paul, MN, USA) consisting of two caliper arms. This measurement tool was reported to be highly reliable for measuring height differences between landmarks [25]. In this study, before measurement, the subjects stood upright with the front of their thigh in contact with a fixed table, then the caliper tip of the PALM was positioned on the ASIS and PSIS on the paralytic side, respectively, to measure the anterior tilt angle. 

The isometric strength of the hip extensor was measured using a handheld dynamometer (Model 01163; Lafayette Inc., IN, USA). The subjects were instructed to stretch their hip joint, flex their knee joints by 90°, and to extend their legs backwards against the hand-held dynamometer for 5 seconds in a side-lying position. The average value of three measurements was used. The handheld dynamometer is known to have high intra-rater and inter-rater reliability in patients with nervous system damage [26]. 

Gait ability was measured using the 10-meter walk test (10MWT). This measurement tool measures the time it takes to walk 10 meters, and the intra-rater and inter-rater reliabilities have been reported to be high [27]. 

### 2.4. Data Analysis

Statistical analysis was performed using SPSS 21.0 (IBM, Armonk, NY, USA). The Shapiro–Wilk test was used to evaluate the normality of the variables. Independent *t*-tests and chi–square tests were used to compare the baseline characteristics of the two groups of continuous and categorical variables. A paired *t*-test was performed to examine the changes in pelvic inclination, strength, and gait speed within a group. An independent *t*-test was used to determine any significant differences between the two groups in the amount of change in pelvic inclination, strength, and gait speed before and after 6 weeks of training. The significance level was set at *p* < 0.05.

## 3. Results

### 3.1. General Characteristics of Participants

No significant difference was found in the general characteristics between the PPTT and control groups before treatment (Table 1). In this study, there was one person who complained of redness due to the tape, but no other person complained of any special side effects.

### 3.2. Changes in Pelvic Inclination

The PPTT group (mean change, each 18.2 ± 3.6 vs. 14.2 ± 2.8, *p* < 0.05) showed a greater degree of improvement in pelvic inclination than the control group (mean change, each 16.9 ± 2.8 vs. 15.5 ± 2.8, *p* < 0.05) (Table 2).

### 3.3. Changes in Muscle Strength

Significantly greater improvements in hip extensor muscle strength were observed in the PPTT group (mean change, each 9.9 ± 2.4, 14.6 ± 3.0, *p* < 0.05) than in the control group (mean change, each 10.7 ± 1.7, 12.6 ± 1.6, *p* < 0.05) (Table 3).

### 3.4. Changes in Gait Speed

After training, the PPTT group (26.0 ± 5.1 vs. 21.1 ± 4.1, *p* < 0.05) showed a more significant improvement in gait speed than the control group (24.6 ± 4.6 vs. 22.3 ± 4.0, *p* < 0.05) (Table 4). 

## 4. Discussion

In this study, we investigated whether the application of posterior pelvic tilt taping in stroke patients with an anterior pelvic inclination affected pelvic inclination. As a result, the anterior pelvic tilt angle in the PPTT group was significantly decreased compared to the control group. Unlike traditional methods, taping applied to the pelvis has elasticity that exceeds the original length [11], and this elasticity increases overall joint movement, skin deformation [28], and stimulation of cutaneous mechanoreceptors [29]. In a study that investigated the effect of taping in lower-back pain patients with increased lordosis, the anterior pelvic tilt was reduced by applying tape to the RA and EO. They reported that as stimulation to mechanoreceptors activates nerve impulses, the strength of the abdominal muscles increases, and consequently, the anterior pelvic tilt decreases [17]. In addition, Bozorgmehr et al. argued that the length–tension relationship of the muscle could be optimized as the application of taping can continuously pull the fascia concentrically and shorten the distance between the muscle origin and insertion, and this could have a positive effect on the joint alignment [18]. The reason the anterior pelvic tilt decreased in this study is thought to be because, as in the previous study, the attachment method, which increases the elasticity of the tape, stimulates the skin’s mechanoreceptors to activate the muscles involved in the posterior tilt of the pelvis, such as the RA and EO. Trunk muscle attaches to the pelvis and provides core stability, which is an important factor in the normal postural alignment of the pelvis [30,31]. Since the trunk is bilaterally innervated and connected by the linea alba with fascia, damage to one side of the brain affects all of the abdominal muscles, which in turn affects the position of the anterior superior iliac spines [9]. The normal anterior pelvic tilt angle was reported as 11 ± 4° [22,23,24]. In this study, a stroke patient with a pelvic anterior tilt angle of 15° or more was targeted. After training, the anterior pelvic tilt angle of the PPTT group was 14.2°, which was within the normal range, indicating clinically significant results.

In addition, in this study, as a result of examining the effect on the muscle strength of the hip extensor involved in the posterior tilt of the pelvis after training, it was confirmed that the PPTT group had a significant improvement compared to the control group. In stroke patients, altered pelvic alignment makes trunk–pelvic dissociation difficult or decreases control of the hip muscles around the pelvis, resulting in asymmetrical weight distribution on the affected side during gait [32], which in turn leads to a weakening of the affected limb due to non-use. In a study on the pelvic alignment of stroke patients, pelvic asymmetry was reported to have a significant correlation with weight-bearing asymmetry [9]. In addition, Verheyden et al. reported that the posture of stroke patients is forward leaning compared to healthy adults, and that posture control ability decreased significantly as the bent increased [10]. In this study, in the case of the PPTT group, posterior pelvic tilt taping was applied during sitting-to-standing, indoor walking, and stair walking training. Although the change in the center of mass was not measured in this study, it is thought that as the alignment of the pelvis was improved, the center of mass was moved backward, thereby promoting the muscle activity of the hip extensor. Dubey et al. reported that the hip extensor’s muscle strength and gait speed were improved after pelvic stability training, which is related to the recruitment of more motor units and the reorganization of the muscle fiber structure as an adaptive response to postural alignment [1]. 

Previous studies reported that stroke patients exhibit low temporal synchronization between the pelvis and lower extremities when walking or performing functional postures and showed anterior pelvic tilt with impaired motor function [5,6]. Moreover, Kim et al. confirmed that the anterior pelvic tilt in stroke patients had significant negative correlations with gait velocity, cadence, and step length [22]. In addition, in stroke patients, weakening of the hip extensor, which is most involved in the terminal stance of the gait, affects the movement control of the hip and knee flexion during the ipsilateral swing phase, resulting in a decrease in walking speed [33,34]. In this study, the gait speed of the PPTT group significantly increased after training compared to the control group, which is thought to be due to the improvement of the anterior pelvic tilt and the resulting increase in the muscle strength of the hip extensor, resulting in an improved posture in the stance phase.

In this study, we confirmed that posterior pelvic tilt taping has a significant effect on pelvic inclination, muscle strength, and gait ability in stroke patients. Pelvic asymmetry causes a negative effect on postural control in stroke patients, but there is no exercise specifically designed for this, and it is more difficult to correct the patient. If the taping attachment method used in this study is used during weight support training, which is widely used for the rehabilitation of stroke patients, it will help to correct the pelvic alignment easily at home by activating the muscles that posteriorly tilt the pelvis. However, it is difficult to generalize due to the small number of subjects, and it was not confirmed whether activation of the trunk muscle was increased, or the center of gravity was changed. Therefore, in a future study, it will be necessary to examine the effects on various gait variables such as cadence, step length, and gait symmetry, as well as the angle of the trunk and lower limb joints whilst walking using a 3D motion analyzer, along with the activation of the trunk muscles.

## 5. Conclusions

The results of this study demonstrated that posterior pelvic tilt taping can improve pelvic tilt, leg muscle strength, and gait in stroke patients with excessive anterior pelvic tilt.

## Figures and Tables

**Figure 1 jcm-10-02381-f001:**
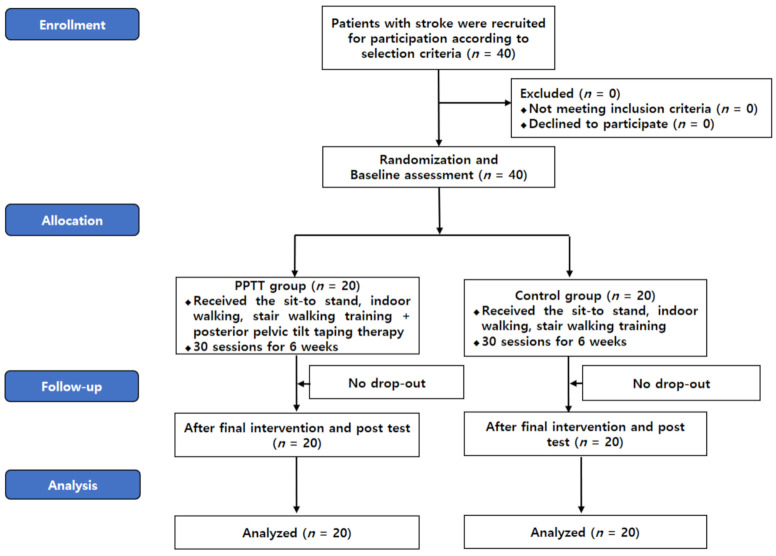
CONSORT flow diagram.

**Figure 2 jcm-10-02381-f002:**
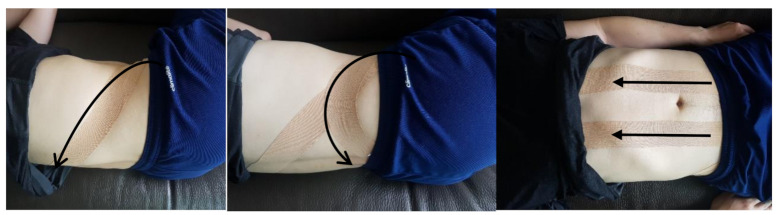
Application of the posterior pelvic tilt taping.

**Table 1 jcm-10-02381-t001:** Common and clinical characteristics of the subjects (N = 40).

Variables	PPTT Group (*n* = 20)	Control Group (*n* = 20)	*p*
Sex (male/female)	14/6	15/5	0.723 ^b^
Affected side(right/left)	12/8	11/9	0.749 ^b^
Age (years)	55.8 ± 8.5 ^a^	54.4 ± 9.9	0.361 ^c^
Height (cm)	165.9 ± 9.1	166.5 ± 9.9	0.856 ^c^
Weight (kg)	62.4 ± 8.7	63.9 ± 8.4	0.758 ^c^
Stroke duration (months)	8.0 ± 1.9	7.1 ± 2.6	0.315 ^c^
Disease (diabetes/hypercholesterolemia/hypertension/≥ 2)	1/1/12/6	1/2/10/7	
Work (engineer/white-collar jobs/etc.)	5/10/5	6/8/6	
Sport activities (none/running/golf/tennis)	14/3/3/2	12/4/3/1	

PPTT, Posterior pelvic tilt taping. ^a^ mean ± standard deviation, ^b^ chi–square test, ^c^ independent *t*-test.

**Table 2 jcm-10-02381-t002:** Changes in pelvic inclination of the study participants (N = 40).

	PPTT Group			Control Group		Difference	*p*-Value
	Pre-Test	Post-Test	Difference	Pre-Test	Post-Test		
Pelvic inclination (°)	18.2 ± 3.6	14.2 ± 2.8 *	−4.0 ± 2.8	16.9 ± 2.8	15.5 ± 2.8 *	−1.4 ± 1.2	<0.001

PPTT, Posterior pelvic tilt taping. * Significant difference between pre-test and post-test (*p* < 0.05).

**Table 3 jcm-10-02381-t003:** Changes in the muscle strength of the study participants (N = 40).

	PPTT Group			Control Group		Difference	*p*-Value
	Pre-Test	Post-Test	Difference	Pre-Test	Post-Test		
Muscle strength (kg)	9.9 ± 2.4	14.6 ± 3.0 *	4.7 ± 2.3	10.7 ± 1.7	12.6 ± 1.6 *	1.8 ± 1.7	<0.001

PPTT, Posterior pelvic tilt taping. * Significant difference between pre-test and post-test (*p* < 0.05).

**Table 4 jcm-10-02381-t004:** Changes in the gait speed of the study participants (N = 40).

	PPTT Group			Control Group		Difference	*p*-Value
	Pre-Test	Post-Test	Difference	Pre-Test	Post-Test		
10MWT (sec)	26.0 ± 5.1	21.1 ± 4.1 *	−4.9 ± 1.6	24.6 ± 4.6	22.3 ± 4.0 *	−2.3 ± 2.8	0.003

PPTT, Posterior pelvic tilt taping; 10MWT, 10 m walk test. * Significant difference between pre-test and post-test (*p* < 0.05).

## Data Availability

The data presented in this study are available on request from the corresponding author.

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
