# Peer review of "Effect of Posterior Pelvic Tilt Taping on Pelvic Inclination, Muscle Strength, and Gait Ability in Stroke Patients: A Randomized Controlled Study"

_jcm, 2021, doi:10.3390/jcm10112381_

Round 1
Reviewer 1 Report
Dear all
I realize that authors have many journals to consider when they want to publish their work, so I appreciate your interest in JCM; I am very sorry not to be able to write in a more positive way. It is evident that you have put a great deal of effort into this project and I want to praise your efforts. The paper is well written but needs further edits. Unfortunately, the actual contribution from your study is not clear or strong. The manuscript as currently written suggests that it might be suitable for sharing information about this topic, but the data that you reported is not representative to state with certainty your conclusions. I should like to thank you for give me an opportunity to consider this work for publication. It may be that the you would like to consider resubmitting it, in which case I hope that the comments from my review may help you to revise it before resubmitting it. These comments are given below.
Best Regards
- Introduction section: is too poor; references are missing in few sentences; clearly describe the rationale for this study of yours and how this would change the current clinical practice for the better to help patients;
- Experimental Section: 2.1. Participants: you should provide additional information about patients (previous pathologies, diabetes, hypercholesterolemia, smoking, sports, work, etc., and all information useful for profiling patients, so that detailed information can be given to other clinicians)
- discussion section: Discussions should be reviewed in light of the overall improvement of the paper. Redundant sentences and prewritten information should be avoided. Focus on take-home messages and how that information impacts the clinical practice of management these patients
- the paper will improving if you insert the images of intervention: taping procedure;
- Reference section: is too poor probably.
Author Response
Comments from Reviewer 1:
I realize that authors have many journals to consider when they want to publish their work, so I appreciate your interest in JCM; I am very sorry not to be able to write in a more positive way. It is evident that you have put a great deal of effort into this project and I want to praise your efforts. The paper is well written but needs further edits. Unfortunately, the actual contribution from your study is not clear or strong. The manuscript as currently written suggests that it might be suitable for sharing information about this topic, but the data that you reported is not representative to state with certainty your conclusions. I should like to thank you for give me an opportunity to consider this work for publication. It may be that the you would like to consider resubmitting it, in which case I hope that the comments from my review may help you to revise it before resubmitting it. These comments are given below.
Response:
- First of all, all authors thank you for evaluating our work. Your comments have improved the quality of our paper, and we thank you for this again.
- We modified the content of the entire manuscript by referring to the valuable points of the Reviewer. We hope that our efforts will satisfy you.
Introduction section:
Comment#1. is too poor; references are missing in few sentences; clearly describe the rationale for this study of yours and how this would change the current clinical practice for the better to help patients.
Response:
- We deeply appreciate your meticulous pointed outs. In view of your comments, we reinforced the content of the Introduction section.
- First, it has been revised so that the content is more clearly expressed in the sentence with the unclear content.
- Second, the references were inserted in the following two sentences where no references were provided.
- ‘Lee et al. reported that a significant difference occurred in the pelvic inclination angle when seated workers were divided between a group in which anterior pelvic tilt taping was applied and a group in which it was not, and maintained a slumped sitting posture for 30 min [16].’
- ‘In a study in which posterior pelvic tilt taping was applied for a week to patients having lower-back pain with hyperlordosis, it was found that lumbar lordosis, pain, disability, and abdominal thickness were improved compared with the control group [18].’
- Third, taping was defined and background for intervention was additionally presented.
- With the consent of all authors, we hope that the revised content will meet your standards.
Experimental section:
Comment#1. 2.1. Participants: you should provide additional information about patients (previous pathologies, diabetes, hypercholesterolemia, smoking, sports, work, etc., and all information useful for profiling patients, so that detailed information can be given to other clinicians).
Response:
- All authors would like to thank you for your good point. The information you pointed out is helpful information for other clinicians, and we additionally presented information on other diseases that the subjects have, excluding stroke, in the Table 1. In addition, we also presented additional information related to the subjects' leisure and sports activities in the Table 1. We hope that the information we have presented is satisfactory to you.
Discussion section:
Comment#1. Discussions should be reviewed in light of the overall improvement of the paper. Redundant sentences and prewritten information should be avoided. Focus on take-home messages and how that information impacts the clinical practice of management these patients.
Response:
- All authors are deeply grateful for your good points. We agree with your comments, and in order to revise these, we rewrote the contents of the Discussion section. We additionally described clinical messages for other clinicians, along with interpretation of the study results.
Comment#2. the paper will improving if you insert the images of intervention: taping procedure.
Response:
- As per your suggestion, we presented additional photos for the posterior pelvic tilt taping. In addition, the content of this is additionally mentioned in the Experimental section.
Reference section:
Comment#1. is too poor probably.
Response:
- We described the contents by using more references in the Introduction and Discussion sections. Along with the use of additional references, the content of the Discussion section has also been additionally described and reinforced. A total of 34 references were used, but if you would like to use more references, we will correct them again.

Reviewer 2 Report
It is a well structured document and aligned with the scope of the journal. Minor comments have been added that could be integrated.

Author Response
Comments from Reviewer 2:
Comment#1. (Page 1) This parragraph must be the first one of the itroducction.
‘The pelvis is an important structure that connects the torso and lower limbs to support the weight and transmits the weight to the lower limbs. In addition, the pelvis is a part of the lower trunk in the sitting position but becomes a functional element of the lower limb when standing or walking [5]. Therefore, changes in pelvic alignment in a standing position also affect balance, gait, and functional performance [6].’
Response:
- Thanks for the good suggestion from the Reviewer. All authors agreed that it is better to place this paragraph at the beginning of the Introduction section as you suggest, due to the flow of content.
- We moved the paragraph to the beginning of the Introduction section as you suggested, and added some references to reinforce the content of this paragraph.
Comment#2. (Page 2) please define "Taping" intervention.
Response:
- According to your good point, we defined 'Taping' and also provided additional information about the taping method in the Introduction section.
- ‘Kinesio taping was introduced by Kenzo Kase as a method of attaching an elastic adhesive tape with elasticity similar to that of the skin, and is currently used in rehabilita-tion for various purposes [11].’
Comment#3. (Page 3) Please add a photo if you have one.
Response:
- As per your suggestion, we presented additional photos for the posterior pelvic tilt taping. In addition, the content of this is additionally mentioned in the Experimental section.
Comment#4. (Page 4) ‘The isometric strength’. Please specify if only one attempt was made, if there were several or how long the contraction was maintained.
Response:
- Participants maintained muscle contraction for 5 seconds to measure the isomeric strength, and a total of 3 measurements were taken and averaged. These are additionally described in the '2.4. Outcome measurements' section.
Comment#5. (Page 4) ‘extend their legs backwards against the hand-held dynamometer in a sidelying position’. the participants were lying down or standing at the time of the test.
Response:
- We thank you for your good point. In general, in healthy people, muscle strength measurements of the hip extensor are performed in a standing position. However, it is difficult for stroke patients to maintain a standing posture due to paralysis, and this may cause compensation in other parts of the body. In order to accurately evaluate only the subject's hip extensor, we performed the evaluation in a side-lying position that can minimize the load of gravity and compensation of other parts of the body.
- We additionally describe this content in the '2.4. Outcome measurements' section.
Comment#6. (Page 4) ‘CSS (score)’ please add in the table legend the meaning of CSS.
Response:
- We modified 'CSS' to 'Pelvic inclination (°)' for more appropriate notation.
Comment#7. (Page 5) ‘the center of mass was moved backward,’. do you measure it? Please make a relation between this and functional task.
Response:
- We thank you for your sharp point. In a study on pelvic alignment in stroke patients, it was reported that pelvic asymmetry had a significant correlation with weight-bearing asymmetry. Unfortunately, although the shift of the center of mass was not measured in this study, the hypothesis was explained that the center of mass would have moved backward because the anterior tilt angle of the pelvic decreased after the intervention, which in turn promoted the muscle activity of the hip extensor.
- The related research contents were described as follows, and since the actual movement of the center of mass was not measured, these contents were additionally described as a limitation of the study in the Discussion section.
- ‘In a study on the pelvic alignment of stroke patients, pelvic asymmetry was reported to have a significant correlation with weight-bearing asymmetry [9]. In addition, Verheyden et al. reported that the posture of stroke patients is forward leaning posture compared to healthy adults, and that posture control ability decreased significantly as the bent in-creased [10]. In this study, in the case of the PPTT group, posterior pelvic tilt taping was applied during sit-to-standing, indoor walking, and stair walking training. Although the change in center of mass was not measured in this study, it is thought that as the alignment of the pelvis was improved, the center of mass was moved backward, thereby promoting the muscle activity of the hip extensor. Dubey et al. reported that the hip extensor's muscle strength and gait speed were improved after pelvic stability training, which is related to the recruitment of more motor units and the reorganization of the muscle fiber structure as an adaptive response to postural alignment [1].’
